Parental emotional neglect and academic procrastination: the mediating role of future self-continuity and ego depletion

Ma Chensen 1
Song Jingjing jingsong321@126.com 2
1 Tongji Hospital, Tongji Medical College, Huazhong University of Science and Technology , Wuhan , China
2 Institute of Applied Psychology, China University of Geosciences , Wuhan , China
Cloninger C Robert
Electronic publication date: 2023 Oct 23
Publication date: 2023
Volume: 11
Electronic Location ID: e16274
Received 2023 May 25; Accepted 2023 Sep 20
Copyright: ©2023 Ma and Song
Copyright year: 2023
Copyright holder: Ma and Song
License: This is an open access article distributed under the terms of the Creative Commons Attribution License, which permits unrestricted use, distribution, reproduction and adaptation in any medium and for any purpose provided that it is properly attributed. For attribution, the original author(s), title, publication source (PeerJ) and either DOI or URL of the article must be cited.
License URL: https://creativecommons.org/licenses/by/4.0/

Keywords: Academic procrastination, Future self-continuity, Ego depletion, Parental emotional neglect

Funding: The National Natural Science Foundation of China (Young Scholar Grant) 31800941 Open Fund from Key Research Institute of Humanities and Social Sciences in Hubei Province-Research Center of University Student Development and Innovation Education, a guided program DXS2023022 This research was supported by grants from the National Natural Science Foundation of China (Young Scholar Grant31800941) and the Open Fund from Key Research Institute of Humanities and Social Sciences in Hubei Province-Research Center of University Student Development and Innovation Education, a guided program grant (DXS2023022). The funders had no role in study design, data collection and analysis, decision to publish, or preparation of the manuscript.

==============================
The purpose of this study was to explore the effect of parental emotional neglect on the academic procrastination of late adolescents and further analyze the mediating role of future self-continuity and ego depletion. This study included 609 college students, 344 males and 265 females, ranging in age from 17 to 21 years (M = 18.39, SD = 0.82), who responded to four questionnaires measuring parental emotional neglect, academic procrastination, future self-continuity, and ego depletion, respectively. The results showed that future self-continuity and ego depletion mediated the association between parental emotional neglect and late adolescents’ academic procrastination in a serial pattern. Perceived higher levels of parental emotional neglect correlated with lower future self-continuity and higher ego depletion in these late adolescents, leading to higher levels of academic procrastination.

Introduction

Academic procrastination reflects a failure of self-regulation, or an individual’s failure to complete the learning-related behaviors, tasks, and activities that they intend to complete within a predetermined time (Klingsieck, 2013; Wang et al., 2021). Academic procrastination is a universal problem, with previous research showing that 52% of students reported delaying their academic tasks (Ozer, Demir & Ferrari, 2009). Academic procrastination has also been shown to be positively associated with many maladaptive outcomes, such as lower levels of self-esteem, more feelings of anxiety, poorer academic performance (Steel & Ferrari, 2013), higher levels of problematic mobile phone use (Hong et al., 2021), and higher dropout intentions (Scheunemann et al., 2022). This study focused on academic procrastination in late adolescents and explored the factors influencing academic procrastination. According to ecological systems theory, family environment is the most important environmental factor affecting childhood development (Bronfenbrenner, 1979). Current study intended to explore the relationship between parental emotional neglect and academic procrastination and further analyze its mechanism. The results of the current study show how parental emotional neglect affects academic procrastination, and may be beneficial for finding targeted interventions to reduce the academic procrastination of students.

Parental emotional neglect and academic procrastination

Many studies have explored the reasons behind academic procrastination. Individual variables such as self-regulation, self-efficacy, self-esteem, and fear of failure all influence academic procrastination (Li, Gao & Xu, 2020; Ma et al., 2022; Wang et al., 2021; Zarrin, Gracia & Paixão, 2020) Meanwhile, negative emotional states, such as anxiety, depression, and worry are all positively associated with academic procrastination (Gadosey et al., 2021; Rahimi & Vallerand, 2021). Furthermore, task-related demands (such as task was incomplete, vague, or ambiguous) also correlate to academic procrastination (Wieland et al., 2022).

Ecological systems theory indicates that family environment is the most important environmental factor affecting childhood development (Bronfenbrenner, 1979). Previous studies have shown that positive parent–child relationships and parental autonomy support are both positively associated with school engagement (Havermans, Sodermans & Matthijs, 2015) and academic success (Vasquez et al., 2015). Some studies have also found that a lack of family education, low parent–child emotional connection, and high paternal alienation lead to a high possibility of academic procrastination in children (Chen, 2017; Woo & Yeo, 2019). Parental emotional neglect refers to parents ignoring the basic emotional needs of children (Deng et al., 2007; Salokangas et al., 2020). Parental emotional neglect is a key factor in low parent–child emotional connection and high parental alienation. We hypothesized that parental emotional neglect would be positively associated with academic procrastination in late adolescents.

Parental emotional neglect is not as immediately evident as other forms of parental abuse, such as verbal or physical abuse. Its negative effects are also not immediately apparent and gradually intensify over time. No previous research has directly explored the relationship between parental emotional neglect and academic procrastination. The purpose of the current study was to explore the effect of parental emotional neglect on the academic procrastination of late adolescents and further analyze the mechanisms behind this relationship.

The mediating role of future self-continuity

The decision to learn without procrastination requires two conditions: (1) motivation to learn and (2) a learning plan. Future self-continuity refers to the extent to which individuals feel connected to and compatible with their future selves (Rutchick et al., 2018). Individuals with high future self-continuity are aware that their current behaviors affect their future self-development, and thus have high learning motivation. High levels of self-control can ensure the implementation of a learning plan. Ego depletion refers to people who have lost self-control after activities that consume self-control resources (Hagger et al., 2010), and correlates with academic procrastination. Therefore, it is necessary to pay attention to the mediating role of future self-continuity and ego depletion simultaneously in the relationship between parental emotion neglect and academic procrastination.

We suspected future self-continuity would play a mediating role in the relationship between parental emotion neglect and academic procrastination. To our knowledge, no previous study has directly analyzed the relationship between parental emotional neglect and future self-continuity, but some have studied the relationship between parental emotional neglect and self-cognition and self-evaluation. These previous studies demonstrated that parental emotional neglect leads to self-denial, self-doubt, and low self-esteem (Shah et al., 2021; Yu & Liu, 2020). These negative evaluations of present self would lead to less thoughts about the future and higher levels of hopelessness (Ferrari et al., 2012; Yu & Liu, 2020). Furthermore, parental emotional neglect also leads to more negative emotions in adolescents. Individuals who experience greater negative affective states may be less likely to include the pattern of future self’s goals, affect, and thoughts into present self, as negative affective states decrease mental flexibility and cognitive broadening (Fredrickson & Joiner, 2002). We hypothesized that parental emotional neglect would be negatively associated with the future self-continuity of late adolescents.

High future self-continuity helps individuals experience their future self as a direct extension of their present self, which is important for guiding appropriate emotional responses and daily goal-oriented behaviors (Damasio, 2010). Future self-continuity improves decision-making (Sani, 2008), workplace proactivity (Strauss & Parker, 2015), and action motivation (Lewis & Oyserman, 2015). Individuals who do not feel connected to their future self are more likely to make immediate choices that have negative long-term consequences. Discontinuities within the temporal sense of self can result in maladaptive planning (Damasio, 2010), leading to unethical choices, further decreasing overall well-being (Singer & Bluck, 2001).

Future self-continuity has been shown to play an important role in academics, and is negatively associated with academic procrastination (Blouin-Hudon & Pychyl, 2015; Blouin-Hudon & Pychyl, 2016). Adolescents who are academic procrastinators focus on smaller, immediate rewards and think less about long-term consequences (Sirois & Pychyl, 2013). We hypothesized that future self-continuity would be negatively associated with academic procrastination, and that future self-continuity would play a mediating role in the relationship between parental emotional neglect and academic procrastination.

The mediating role of ego depletion

We also hypothesized that ego depletion would play a mediating role in the relationship between parental emotional neglect and academic procrastination. Parental emotion neglect will lead to ego depletion. Firstly, parental emotional neglect triggers a stress response, consuming psychological resources, leading to low self-esteem and a low sense of hope (Shah et al., 2021; Yu & Liu, 2020). Negative self-evaluation, self-perception, and the absence of positive psychological qualities lead to ego depletion (Rahman et al., 2019; Tan et al., 2012). Secondly, parental emotional neglect also induces negative emotions, such as depression and anxiety, which are positively associated with high levels of ego depletion (Shmueli & Prochaska, 2012). Furthermore, poor interpersonal relationships also consume individual self-control resources (Bertrams & Pahl, 2014). Previous studies have found that participants who perceived social rejection performed worse on binaural listening tasks than participants who perceived social acceptance, confirming that social rejection leads to ego depletion (Baumeister et al., 2005).

Ego depletion also likely increases academic procrastination. Ego depletion, as the lack of sufficient self-control resources, is positively associated with declines in the ability to self-control. Self-control is a psychological resource or tool that helps individuals make progress toward their desired future self. Previous research has found that self-control is a robust predictor of academic performance (e.g., (Duckworth, Quinn & Tsukayama, 2012). Moreover, ego depletion also promotes risk-taking, immediate gratification rather than long-term gains, and induces a variety of maladaptive behaviors such as addiction (Muraven & Baumeister, 2000) and violent behavior (Tan et al., 2012). Therefore, we assumed that ego depletion correlated to academic procrastination.

The relationship between future self-continuity and ego depletion

We hypothesized that future self-continuity would be negatively associated with ego depletion. Individuals with high future self-continuity are more likely to consider the impact of their current behavior on their future selves, directing focus toward future goals and away from short-term demands (Hershfield, Cohen & Thompson, 2012), increasing the importance of future success. The consideration of future consequences leads people have more willingness to drain the resource, further enables individuals to effectively exert self-control and increases the behavioral regulation that facilitates the attainment of future outcomes (Adelman et al., 2016; Hershfield, Cohen & Thompson, 2012; Jiang et al., 2022), leading to less ego depletion.

Research has shown the negative relationship between future self-continuity and ego depletion. One previous study found that ego depletion can be circumvented by increasing self-awareness; no decrease in performance was observed for depleted participants who were exposed to a self-awareness prime (Alberts, Martijn & Vries, 2011). Muraven & Slessareva (2003) found that initially depleted individuals persevered longer at a self-control task if they believed that their persistence in the task would be beneficial to themselves or others. Thus, we assumed that the future self-continuity was negatively associated with ego depletion.

Current study

Although some studies have explored the influence of parental relationship on academic procrastination, no study directly focused on the influence of parental emotional neglect on academic procrastination in late adolescents. However, the effect of parental emotional neglect is always hidden, not easy to detect, and far-reaching. Moreover, it is necessary to pay attention to the influence of parental emotional neglect on academic procrastination in the context of Chinese culture. Because the parents of today’s late adolescents face intense social pressure and were raised when material resources were scarce, they are more likely to only focus on the material and educational needs of their children and ignore emotional care. Emotional neglect is common among Chinese parents. One study investigated 1,041 Chinese students, and 76.46% of them claimed they suffered emotional neglect in childhood (Sun, Liu & Yu, 2019). In addition, it is necessary to further explore the mechanism of parental emotional neglect affecting academic procrastination. From two perspectives: motivation to learn and a learning plan, current study focused on the mediating role of future self-continuity and ego depletion.

The current study focused on Chinese college students and explored the positive relationship between parental emotional neglect and academic procrastination, and further analyzed the mediating role of future self-continuity and ego depletion. This study used a serial mediation model with two mediators and three indirect pathways: through future self-continuity, through future self-continuity and ego depletion, and through ego depletion. We hypothesized that parental emotional neglect would lead to low levels of future self-continuity in late adolescents, resulting in increased ego depletion and higher levels of academic procrastination (see Fig. 1).

Figure 1 The hypothesized model of parental emotional neglect affecting academic procrastination.

The specific hypotheses tested in this study were: that parental emotional neglect is positively associated with academic procrastination in late adolescents (H1); future self-continuity plays a mediating role in the relationship parental emotional neglect and academic procrastination (H2), parental emotional neglect is negatively associated with future self-continuity (H2a), and future self-continuity is negatively associated with academic procrastination (H2b); ego depletion also plays a mediating role in the relationship between parental emotional neglect and academic procrastination (H3), parental emotional neglect is positively associated with ego depletion (H3a), and ego depletion is positively associated with academic procrastination (H3b); and that future self-continuity and ego depletion mediate the relationship between parental emotional neglect and academic procrastination in a serial pattern (H4), and future self-continuity is negatively associated with ego depletion (H4a).

Method

Participants

Participants were recruited for a series of studies on adolescents development, including negative effect of harsh parenting (Ma & Song, 2023). A total of 620 college students at a university in Central China volunteered to participate. Data from 11 participants were excluded because of invalid responses, leaving 609 total participants. There were 344 males (56.5%) and 265 females (43.5%), with ages ranging from 17 to 21 years (M = 18.39, SD = 0.82). Of the 609 participants, 227 participants lived in in rural areas (37.3%) and 382 in urban areas (62.7%), and 42 participants thought their families were very poor (6.9%), 90 reported their families were a little poor (14.8%), 437 were of average wealth (71.8%), 37 reported their families were a little rich (6.1%), and three reported their families were very rich (0.5%).

Research design

Permission to conduct the research was acquired from the ethics committee of the Institution of Psychology, China University of Geosciences (Wuhan). A questionnaire survey method was used to conduct the research, and parental emotional neglect, ego depletion, future self-continuity, and academic procrastination were all measured. A serial mediation model was analyzed using parental emotional neglect as the independent variable, academic procrastination as the dependent variable, and future self-continuity and ego depletion as mediating variables.

Measurements

Parental emotional neglect

Parental emotional neglect was measured using the emotional neglect subscale from the psychological abuse and neglect scale (Deng et al., 2007). Participants responded to eight statements (e.g., “When I am sad or afraid, my parents don’t comfort me”) on a five-point Likert scale (1 = little or none, 5 = the most). Higher scores indicated higher levels of parental emotional neglect. This scale had good internal consistency (Cronbach’s alpha = 0.89, Composite Reliability = 0.92) and good convergent validity (Average Variance Extracted = 0.58) in current sample.

Ego depletion

Ego depletion was measured using the ego depletion scale (Lanaj, Johnson & Barnes, 2014; Johnson & Joanna, 2015). The validity of this scale was previously demonstrated in a Chinese sample (Ding, Zhang & Zhou, 2020). Participants responded to five statements (e.g., “I feel exhausted”) on a five-point Likert scale (1 = disagree at all, 5 = strongly agree). Higher scores indicated higher levels of ego depletion. This scale had good internal consistency (Cronbach’s alpha = 0.90, Composite Reliability = 0.93) and a good convergent validity (Average Variance Extracted = 0.72) in current sample.

Future self-continuity

One item was used to measure future self-continuity. Participants were told that there were two circles, one representing the present self, and the other representing the future self in ten years. Participants then had a choice between seven pairs of circles with varying degrees of overlap. More overlap between the two circles indicated a closer connection between the future self and present self, and higher levels of future self-continuity. This measurement of future self-continuity has been widely used in previous research (Bartels & Urminsky, 2015; Liu et al., 2018). It has also been demonstrated that single-item measures have acceptable reliability and validity compared with multi-item measures (Wei & Zhang, 2019).

Academic procrastination

Academic procrastination was assessed using the academic procrastination scale conducted by Zheng (2009). Participants answered 17 items covering four aspects of academic procrastination: lack of a learning plan, poor learning status, delayed learning behavior and insufficient learning execution. Items were rated on a five-point Likert scale (1 = little or none, 5 = the most). Higher scores indicated higher levels of academic procrastination. This scale has been widely used in previous research (Song et al., 2016). This scale had good internal consistency in current sample (Cronbach’s alpha = 0.90, Composite Reliability = 0.90). The results of exploratory factor analysis (EFA) showed that all factor load λ > 0.4. This scale also had acceptable convergent validity (Average Variance Extracted values of integrated and isolated factors = 0.38, 0.49, 0.48, 0.43, 0.45).

Procedure

Data were collected as previously described in Ma & Song (2023). After the experimenter described the purpose and content of the survey, participants voluntarily signed informed consent forms and scanned a QR code to obtain the online questionnaire. The questionnaire took approximately ten minutes to complete and measured parental emotional neglect, ego deletion, future self-continuity, and academic procrastination. Other variables, such as harsh parenting, life satisfaction, negative coping style, were also measured, but were not analyzed in the current study.

Statistical analyses

Correlations between parental emotional neglect, future self-continuity, ego depletion, and academic procrastination were calculated using SPSS 22.0 (SPSS Inc. Chicago, IL, USA) with Pearson’s product-moment correlation coefficient. The mediating role of future self-continuity and ego depletion in the relationship between parental emotional neglect and academic procrastination was calculated with the structural equation model using the Amos statistical package, version 7.0.

Results

Pearson correlation coefficient results

The correlations among parental emotion neglect, future self-continuity, ego depletion, and academic procrastination were calculated using the Pearson’s product-moment correlation coefficient (see Table 1). The results showed that parental emotional neglect was negatively associated with future self-continuity, and positively associated with ego depletion and academic procrastination (see Table 1). Participants with higher levels of perceived parental emotional neglect had lower levels of future self-continuity and higher levels of ego depletion and academic procrastination.

Table 1 Pearson correlation coefficient results for parental emotional neglect, future self-continuity, ego depletion, and academic procrastination.

	1	2	3	4	5	6	
1. Gender	–						
2. Age	0.09*	–					
3. Parental emotion neglect	0.12**	0.06	–				
4. Self-continuity	−0.07	−0.01	−0.19***	–			
5. Ego depletion	0.09*	0.06	0.35***	−0.18***	–		
6. Academic procrastination	0.12**	0.07	0.21***	−0.09*	0.49***	–	
M	0.56	18.39	15.36	3.54	13.01	43.09	
SD	0.50	0.82	6.03	1.60	4.79	9.78	
Notes.

Gender is a dummy variable, with male = 1, and female = 0. Mean for gender is the percentage of male students.

* p < 0.05.

** p < 0.01.

*** p < 0.001.

Serial mediation model results

We anticipated that future self-continuity and ego depletion would mediate the relationship between parental emotional neglect and academic procrastination in a serial pattern. To test this hypothesis, we performed structural equation model using the Amos statistical package, version 7.0. Bootstrapping with 1,000 iterations was used to generate an approximation of the sampling distribution in order to obtain confidence intervals that are more accurate than confidence intervals using standard methods (Hayes & Preacher, 2010).

The hypothesized model provided a good fit to the data (χ2 = 443.49, df = 130, comparative fit index (CFI) = 0.95, incremental fit index (IFI) = 0.95, normed fit index (NFI) = 0.93, goodness-of-fit index (GFI) = 0.92, root mean square error of approximation (RMSEA) = 0.06). Table 2 and Fig. 2, which display the standardized estimates of this model, show parental emotional neglect was negatively associated with future self-continuity in late adolescents (β = −0.20, p < 0.001). Parental emotional neglect was positively associated with ego depletion (β = 0.35, p < 0.001), and future self-continuity was negatively associated with ego depletion (β = −0.11, p = 0.01). Ego depletion was positively associated with academic procrastination (β = 0.56, p < 0.001), but neither parental emotional neglect nor future self-continuity were significantly correlated with academic procrastination (β = 0.04, p = 0.33; β = 0.01, p = 0.85). These results support hypotheses H2a, H3a, H3b, H4a, but hypotheses H1 and H2b were not supported by the results of this study.

Table 2 The mediating role of future self-continuity and ego depletion in the relationship between parental emotional neglect and academic procrastination.

Regression equation	Regression coefficient	
Dependent variable	Independent variable	B	LLCI	ULCI	P	
Self-continuity	Emotion neglect	−0.20	−0.28	−0.11	0.002	
Ego depletion	Emotion neglect	0.35	0.25	0.43	0.002	
	Self-continuity	−0.11	−0.20	−0.02	0.02	
Academic procrastination	Emotion neglect	0.04	−0.05	0.13	0.40	
Self-continuity	0.01	−0.07	0.09	0.83	
Ego depletion	0.56	0.47	0.65	0.002	
Notes.

LLCI is Lower-Level Confidence Interval, ULCI is Upper-Level Confidence Interval, if the confidence interval from LLCI to ULCI not include zero, it indicates regression coefficient is significant.

Figure 2 The mediating role of future self-continuity and ego depletion on the effect of parental emotional neglect on academic procrastination.

The mediation effect analysis was performed with SPSS macro-PROCESS using the methods outlined by Hayes (2012). The results showed that the mediating effect of future self-continuity in the relationship between parental emotional neglect and academic procrastination in late adolescents was not significant (β = −0.001, SE = 0.01, LLCI = −0.03, ULCI = 0.02). However, the mediating role of ego depletion was significant (β = 0.25, SE = 0.04, LLCI = 0.17, ULCI = 0.32), and the serial mediating role of future self-continuity and ego depletion was also significant (β = 0.01, SE = 0.01, LLCI = 0.003, ULCI = 0.03). H2 was not supported, H3 and H4 were supported. These results indicate that parental emotional neglect had an indirect influence on the academic procrastination of late adolescents through future self-continuity and ego depletion.

Discussion

The purpose of the current study was to explore the effect of parental emotional neglect on the academic procrastination of late adolescents. Many studies have explored the reasons for academic procrastination. Self-regulation, self-efficacy, self-esteem, fear of failure, emotions, and personality were the most frequently cited individual variables (Wang et al., 2021; Zarrin, Gracia & Paixão, 2020). Parenting is an important factor in childhood development and in the shaping of individual characteristics (including the above-mentioned self-regulation, self-efficacy, self-esteem, fear of failure, emotions, and personality). Therefore, studying the influence of parental emotional neglect on academic procrastination could help provide a further understanding of the causes of procrastination behavior. Moreover, current study further analyzed the mechanism of parental emotional neglect influencing academic procrastination, explored the mediating role of future self-continuity and ego depletion. The results of this study may help in the creation of targeted methods to reduce academic procrastination.

The results of this study showed that parental emotional neglect was positively associated with academic procrastination, which is consistent with the results of previous research demonstrating that parenting is correlated with academic procrastination (Chen, 2017; Woo & Yeo, 2019). Previous research demonstrated that parental emotional neglect leads to many negative outcomes including: negative coping strategies, a high probability of internet addiction (Yang & Huang, 2010), high levels of suicidal ideation (Yu & Liu, 2020), negative self-cognition, negative emotional experiences (anxiety, depression), poor self-control abilities, and poor time management abilities (Rees, 2008). These risk factors increase the probability of academic procrastination.

We found that future self-continuity played a mediating role in the relationship between parental emotional neglect and academic procrastination. Parental emotional neglect was negatively associated with future self-continuity. Parental emotional neglect leads teenagers to have a more negative self-evaluation, have high levels of rumination to think about their negative situation, have more possibility of emotional distress, likely leading to them to have less psychological resources to think about the future and feel less connected to and compatible with their future selves, or low future self-continuity. We also found that future self-continuity was negatively associated with academic procrastination. Procrastination indicates the inability to regulate immediate mood and behavior, as the primacy of the present self is higher than the goals and rewards of the future self. Procrastination has also been found to be negatively associated with a future time perspective and positively associated with a present hedonistic and present fatalistic time orientation (Ferrari & Dıáz-Morales, 2007).

Ego depletion also played a mediating role in the relationship between parental emotional neglect and academic procrastination in our study. Parental emotional neglect was positively associated with ego depletion, which is consistent with our hypothesis and the results of previous research. Parental emotional neglect consumes psychological resources (Yu & Liu, 2020) and induces negative social relations and negative emotions such as depression and anxiety, likely leading to higher levels of ego depletion (Bertrams & Pahl, 2014; Shmueli & Prochaska, 2012). In addition, consistent with our hypothesis, we found that ego depletion increased academic procrastination. First, ego depletion reflects a lack of self-control, and adolescents with lower levels of self-control are more likely to engage in immediate gratification, have behavior problems such as internet addiction, and couldn’t finish learning tasks on time. Second, ego depletion also reflects a negative emotional state. Teenagers with high levels of ego depletion have higher levels of hopelessness, are less motivated to study and work, and are more likely to procrastinate in school.

Practical significance

The results of this study are beneficial for identifying targeted interventions to reduce student academic procrastination. First, correcting parental emotional neglect should be an important measure to reduce the learning problems of adolescents. Although many family interventions programs have recognized the importance of parental care and love, parental emotional neglect is still common. Therefore, practical work should be done to help parents fully understand the psychological needs of adolescents and the long-term negative influence of parental emotional neglect on individual development. Secondly, adolescents with high future self-continuity have lower levels of academic procrastination. We assumed that make future career planning, apply self-leadership strategy (achieve the self-direction and self-motivation) (Wang et al., 2021), can help people improve future self-continuity, and further lead to less academic procrastination. Finally, ego depletion is also an important reason for academic procrastination. Identifying the causes of ego depletion, improving mental resources, and improving self-control abilities may also be key factors in reducing academic procrastination.

Limitations

This study also has limitations. First, causal inferences cannot be made from the current data. Experimental research or longitudinal research is needed to explore possible causality. Second, this study did not include family income, or the education level and marital status of the parents. Future studies should comprehensively investigate demographic information to help identify the impact of family demographics on parental emotional neglect and academic procrastination. Third, this study used a single measure for future self-continuity. Future studies should use a more structured future self-continuity questionnaire. Moreover, the current study focused on the mediating role of future self-continuity and ego depletion, however, the specific demands and difficulty of the academic task are also important factors of academic procrastination that were not considered in this study. In addition, parental influence may differ between fathers and mothers, with one parent impacting the psychology and behavioral development of adolescents in different ways and with different intensity. This study failed to explore the different effects of paternal emotional neglect and maternal emotional neglect on the academic procrastination of late adolescents.

Conclusion

Future self-continuity and ego depletion mediated the association between parental emotional neglect and academic procrastination in late adolescents in a serial pattern. Perceived higher levels of parental emotional neglect correlated with lower levels of future self-continuity and higher levels of ego depletion, likely leading to higher levels academic procrastination.

Supplemental Information

Supplemental Information 1 Categorical data code

Click here for additional data file.

Supplemental Information 2 Data

Click here for additional data file.

Supplemental Information 3 Test instrument permission

Click here for additional data file.

We acknowledge Li Junnan for participating in data collection.

Additional Information and Declarations

Competing Interests

Author Contributions

Human Ethics

Data Availability

The authors declare there are no competing interests.

Chensen Ma performed the experiments, prepared figures and/or tables, authored or reviewed drafts of the article, and approved the final draft.

Jingjing Song conceived and designed the experiments, performed the experiments, analyzed the data, prepared figures and/or tables, authored or reviewed drafts of the article, and approved the final draft.

The following information was supplied relating to ethical approvals (i.e., approving body and any reference numbers):

This study was conducted in accordance with the Declaration of Helsinki. Approval to conduct the study was obtained from the Ethics committee of the Institution of Psychology, China University of Geosciences (Wuhan), China. Informed consent was obtained from all adult participants included in the study.

The following information was supplied regarding data availability:

The data is available at figshare: Song, Jingjing (2023). Parental emotion neglect and academic procrastination. figshare. Dataset. https://doi.org/10.6084/m9.figshare.24173721.v1.

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
