# Peer review of "Parental emotional neglect and academic procrastination: the mediating role of future self-continuity and ego depletion"

_PeerJ, doi:10.7717/peerj.16274_

## Round 0.1 · original submission · Major Revisions

I agree with each of the three reviewers that your paper is of potential interest. However, each section of the manuscript needs thorough revision, including introduction (prior state of field, objectives, hypotheses, expected significance and challenges), all aspects of methods (subject description with demographics, statistical analysis, measures/procedures), reporting of results including Figures), Results (relations of variables, statistical reporting), and Discussion. Each review has specific recommendations for your consideration, I cannot guarantee acceptance of the Revision, but each reviewer and I feel this could be written in a publishable form. We hope these constructive remarks help you in this task.

Reviewer 1 ·

Basic reporting

1. The introduction part needs to clearly state the innovation of this research. For example, to what extent has the previous related research progressed? Is the innovation of this study in the main effect or the mediation effect?

2. The relationship between future self-continuity and ego depletion requires more logical derivations, not just evidence from a single previous empirical study. In addition, although self-control is related to ego depletion, the two do not seem to be completely equivalent.

3. It is better for the author to add a graph to describe the relationship between variables.

4. Some parts only have references from China, if possible, please add references from more sources.

Experimental design

no comment

Validity of the findings

no comment

Reviewer 2 ·

Basic reporting

This study aims at examining on a large sample of college students the associations between parental emotion neglect and academic procrastination, further investigating the mediator role played by future self-continuity and ego-depletion. I think the authors explore a very relevant research question with several practical implication considering the harsh impact of study’s procrastination on later socio-psychological and emotional adjustment. In particular, the study’s findings are very interest and useful for scientist and experts in the field. In particular the clear and detailed way by which the authors describe previous studies, results and implications it is appreciable; however, I think the manuscript need a general revision for what concern some issue.
The only major issue consists of the statistical analyses. In particular my suggestion is to test the mediation model by implementing a structural equation model with the CInteval significance examination.
Below, are reported my detailed comments about some minor issues.
ABSTRACT
#1 Line 28. Please to report the participants’ mean age and gender’s distribution.
INTRODUCTION
#2 Line 47. Please to indicate in the first overall introductory paragraph the study’s overall aim and research questions and the study’s contribution to the current state of art.
CURRENT STUDY
#3. I suggest to the authors to better write this paragraph by indacting:
1) A summarize of the state of the art;
2) The overall aim of the study;
3) The study’s research question;
4) The study’s specific objectives;
5) The study’s hypotheses.
METHOD
PARTICIPANTS
#4. Please better describe participants by describing their socio-demographic variables such as socio-economic status, income, parents’ education, marital status, ecc..
#5. I further suggest to add a paragraph entitled “Statistical analyses”, which describe the implemented statistical analyses and their aims.

Experimental design

I think the manuscpript fits avery weel with the journal's aoms- The research question is very well defined and also the method is described well-

Validity of the findings

The study's findigns are very relevant, however I think that the statistical methods used by authors could be improved by using more complex and sophisticated statistical analyses. Conclusion are also well stated.

Reviewer 3 ·

Basic reporting

See my comments.

Experimental design

See my comments.

Validity of the findings

See my comments.

Additional comments

Thanks for opportunity to review manuscript entitled ‘‘Parental emotion neglect and academic
procrastination: The mediation role of future self continuity and ego depletion’’ for Peerj Journal. The strength of the manuscript includes examining variables of interest in a country where such studies are scarce. Overall, although the article is generally well written and deserves to be published in this journal some revisions must be made before the publication of the article. Because my main philosophy of reviewing a manuscript as reviewer and sometimes an editor to improve the manuscript and not punishing the authors, I provided very specific and detailed peer review of the manuscript to increase its quality and citation potential. I hope authors of the manuscript may benefit from my review. Necessary revisions reported section by section with the page and line number and when possible with suggestions.
1. A figure representing mediation model required for this article.
2. Authors must transpose Table 1 such that 1. Gender must be down and 1 up.
3. Statical reporting along the manuscript is problematic add a space before and after =.
4ç Statistical analyses section in Method completely missing and must be added.
5. Research design in Method completely missing and must be added.
6. Authors need to add a limitation related using single item scale future self-continuity.
7. In the introduction section authors must give information about importance of their study in their cultural context. Specifically authors must answer ‘‘Why it is important to examine the mediation role of future self-continuity and ego depletion in the relationship between parental emotion neglect and academic procrastination İN Chinese cultural context?’’
8. Procedure section must be under the measures.
9. Composite reliability and McDonalds omega is the same thing in the context of CFA. Authors must remove one of them.
10. Note under the tables must be italic.
11. Authors must correct all reporting as per APA 7 rules especially findings and statistical symbols.
12. Apart from above corrections, the article is well-written.

---

## Round 0.2 · Minor Revisions

You have done excellent job of responding to the substantive issues raised in the review. My only concern is that your use of English language is not consistently gramatically correct. It is understandable but the grammatical errors detract from the quality of your work. For example, the first insertion in the introduction begins with "Base on ecological systems theory, ...". This clearly should be "Based on ecological....". Such errors are abundant and so I would encourage you to contact PeerJ editorial staff for guidance on producing a version with better use of English language. The article does not require further scientific review and once I am informed by the editorial staff that the use of language is adequate, I will accept it.

**Language Note:** The Academic Editor has identified that the English language must be improved. PeerJ can provide language editing services - please contact us at copyediting@peerj.com for pricing (be sure to provide your manuscript number and title). Alternatively, you should make your own arrangements to improve the language quality and provide details in your response letter. – PeerJ Staff

---

## Round 0.3 · accepted · Accept

You have addressed my concerns about language and those of the reviewers, so the article is ready for publication. The editing of the language was excellent.